# Nonequilibrium continuous phase transition in colloidal gelation with short-range attraction

Joep Rouwhorst[1], Christopher Ness[2,3], Simeon Stoyanov [4], Alessio Zaccone [2,5,6 ✉] & Peter Schall [1 ✉]

The dynamical arrest of attractive colloidal particles into out-of-equilibrium structures, known as gelation, is central to biophysics, materials science, nanotechnology, and food and cosmetic applications, but a complete understanding is lacking. In particular, for intermediate particle density and attraction, the structure formation process remains unclear. Here, we show that the gelation of short-range attractive particles is governed by a nonequilibrium percolation process. We combine experiments on critical Casimir colloidal suspensions, numerical simulations, and analytical modeling with a master kinetic equation to show that cluster sizes and correlation lengths diverge with exponents ~1.6 and 0.8, respectively, consistent with percolation theory, while detailed balance in the particle attachment and detachment processes is broken. Cluster masses exhibit power-law distributions with exponents $-3/2$ and $-5/2$ before and after percolation, as predicted by solutions to the master kinetic equation. These results revealing a nonequilibrium continuous phase transition unify the structural arrest and yielding into related frameworks.

[1] Institute of Physics, University of Amsterdam, Science Park 904, Amsterdam 1098 XH, The Netherlands. [2] Department of Chemical Engineering and Biotechnology, University of Cambridge, Cambridge CB3 0AS, UK. [3] School of Engineering, University of Edinburgh, Edinburgh EH9 3FB, UK. [4] Unilever R&D Vlaardingen, Olivier van Noortlaan 120, Vlaardingen 3133 AT, The Netherlands. [5] Department of Physics "A. Pontremoli'", University of Milan, via Celoria 16, Milan 20133, Italy. [6] Cavendish Laboratory, University of Cambridge, Cambridge CB3 0HE, UK. ✉email: az302@cam.ac.uk; P.Schall@uva.nl

Jammed out-of-equilibrium structures forming from attractive particles are ubiquitous in nature and many consumer products. Being metastable solids, they are mechanically rigid structures that typically form at low particle density owing to a space-spanning cluster of aggregated particles. Important examples include attractive colloidal particles that aggregate into system-spanning networks known as gelation[1]. Extensive work has gone into mapping the phase boundary of this transition, unraveling jamming diagrams across a range of volume fractions and particle interaction strength[1–11]. Although much studied for strong attraction, where particles stick irreversibly and open structures form[12–17], and at rather low attraction, typically by colloidal depletion interaction, where phase separation occurs by spinodal decomposition into depletant-rich and depletant-poor phases with subsequent arrest[2], the situation is much less clear for intermediate particle attraction, where the structure forms in a highly out-of-equilibrium process, most relevant to structure formation in biology.

At effective interparticle attractions of many $k_BT$, the thermal energy, the detailed balance in the particle attachment and detachment processes is broken, and the system falls out-of-equilibrium; in this case, a description based on an underlying equilibrium phase diagram may no longer apply. This is demonstrated by a recent experimental study on the intermittent dynamics of colloidal gels[18], revealing a marked asymmetry in the cooperative bonding and de-bonding processes. This regime, which is most relevant for biological network formation, is often modeled by cluster kinetic equations, but there is no theory that describes the formation of these out-of-equilibrium structures and that is able to explain or predict, from first-principles, the fractal dimension of the growing clusters, and its relation to the cluster-mass distribution. Furthermore, most experimental studies in the weakly attractive regime are based on depletion interactions that naturally cause phase separation into depletant-rich and depletant-poor phases, which may yield specific routes to gelation, distinct from those of attractive spheres[10].

Besides the arrested spinodal decomposition scenario[2], also a mechanism based on a double-glass transition, or jamming transition of clusters, has been proposed[19], which has recently received experimental verification in terms of the resulting predictions for the gel elasticity and mesoscale structure[20]. In fact, recent two-dimensional simulations suggest that the jamming of attractive spheres falls into a distinct universality class[21]: the continuous growth of clusters is reminiscent of a continuous phase transition with a diverging length scale, different from the more familiar repulsive jamming. Such framework of critical phenomena has been sought as a possible connection between physical, colloidal, and chemical gelation, as it would offer a unifying description[22,23]. Yet, although equilibrium percolation transitions have been discussed for fluid–fluid and fluid–solid transitions[24,25], used to interpret experimental data[10,26,27], and in theoretical models[28], their validity for systems out-of-equilibrium remains unclear.

Here, we combine experiments on tunable attractive colloids with simulations and analytic kinetic modeling to show that the observed gelation of short-range attractive particles into space-spanning structures shows all hallmarks of a nonequilibrium continuous phase transition. We study cluster growth of particles interacting with an effective critical Casimir attraction, as well as via simulations and analytic solutions to the master kinetic equation; the latter encodes the relevant physics in terms of aggregation and spontaneous breakage of the growing clusters. All approaches uniquely converge to show that the observed short-range attractive colloidal gelation is related to a nonequilibrium second-order phase transition, with critical exponents of cluster growth in agreement with percolation theory.

Analytically, this is supported by solving the master kinetic equation in the limit of single-particle detachment predicting the existence of a critical point and power-law cluster-mass distributions. Both predictions are indeed quantitatively confirmed by the experiments and simulations over a range of attractive strengths. These results open up a new nonequilibrium view on gelation and attractive jammed structures in general, relevant for many natural aggregation processes. Our findings identify this structural arrest as an analogue, mirror-image process of yielding, and suggests unification of yielding and dynamical arrest phenomena into a single framework.

## Results

**Cluster growth and critical scaling**. We use colloidal particles suspended in a sucrose binary mixture of lutidine and water, in which attractive critical Casimir forces arise close to the solvent critical point $T_c = 31.0$ °C. The particles have a radius $r = 1$ μm with a polydispersity of 5% and are suspended at a volume fraction of $\phi \sim 0.12$ in the sucrose binary solvent, which matches their density and refractive index, allowing for observation of assembly deep in the bulk with minimal disturbance by gravity (see Methods). Close to $T_c$, attractive critical Casimir forces cause particle aggregation with an effective attractive potential set by the temperature difference $\Delta T = T_c - T$[29–32]. Previous studies have revealed equilibrium phase transitions from gas to liquid and liquid to solid at low attraction[33–36], as well as colloidal aggregation at higher attraction, which was investigated in microgravity[37–39]. To study gelation, we induce sufficiently strong attractive strength between the particles by jumping from room temperature to $\Delta T = 1.2, 1.0, 0.7$, and $0.5$ °C, corresponding to an attraction increasing from ~3 to $10\,k_BT$. For each attraction, we follow the particle-scale aggregation process in a 108 μm by 108 μm by 40 μm volume using confocal microscopy.

The experiments are complemented with molecular dynamics simulations of an equal mixture of particles with radii $r_a/r_b = 1:1.1$ at volume fraction $\phi = 0.12$. Particles follow the overdamped Langevin equation, interacting through a Mie potential with parameters chosen to match the rather short attractive range of $\approx 0.08 r_a$ of the experiments (see Methods). The attractive strength is given by $\epsilon/k_BT$, where $\epsilon$ is the prefactor of the potential and $k_BT$ is the thermal energy. To test the generality of the computational results, we also perform simulations with a square-well potential, on particles with the same size ratio and effective attraction, as defined from the corresponding second virial coefficients. To compare with the phase behavior of adhesive hard spheres, we compute the Baxter parameter, $\tau$, and find that the onset of gelation we observe at $\tau \sim 0.1$, is in very good agreement with the gelation transition of adhesive hard spheres[10], see Supplementary Note 1 and Supplementary Figures 1 and 2.

Experiments on the aggregating colloidal particles reveal growing clusters, the largest of which eventually spans the field of view (Fig. 1a–c). By plotting the size evolution of the largest and second-largest cluster in Fig. 1d, we identify the onset of space-spanning structures by the sudden increase of the largest cluster, and concomitant decrease of the second-largest cluster, which becomes part of the largest cluster at gelation.

We investigate the onset of gelation by looking at the evolution of the average coordination number $z$, i.e., the average number of bonded neighbors of a particle. This number increases as clusters grow to saturate at a value $z_{max}$, see Fig. 1e. We take $z$ as the order parameter of the gelation transition and plot the fraction of particles in the largest cluster, $f_z$, as a function of $z$ in Fig. 2a. It increases sharply upon approaching the transition, indicating that the largest cluster abruptly absorbs a large number of particles. We find a divergence $f_z \sim (z_c - z)^{-\sigma}$ upon approaching the

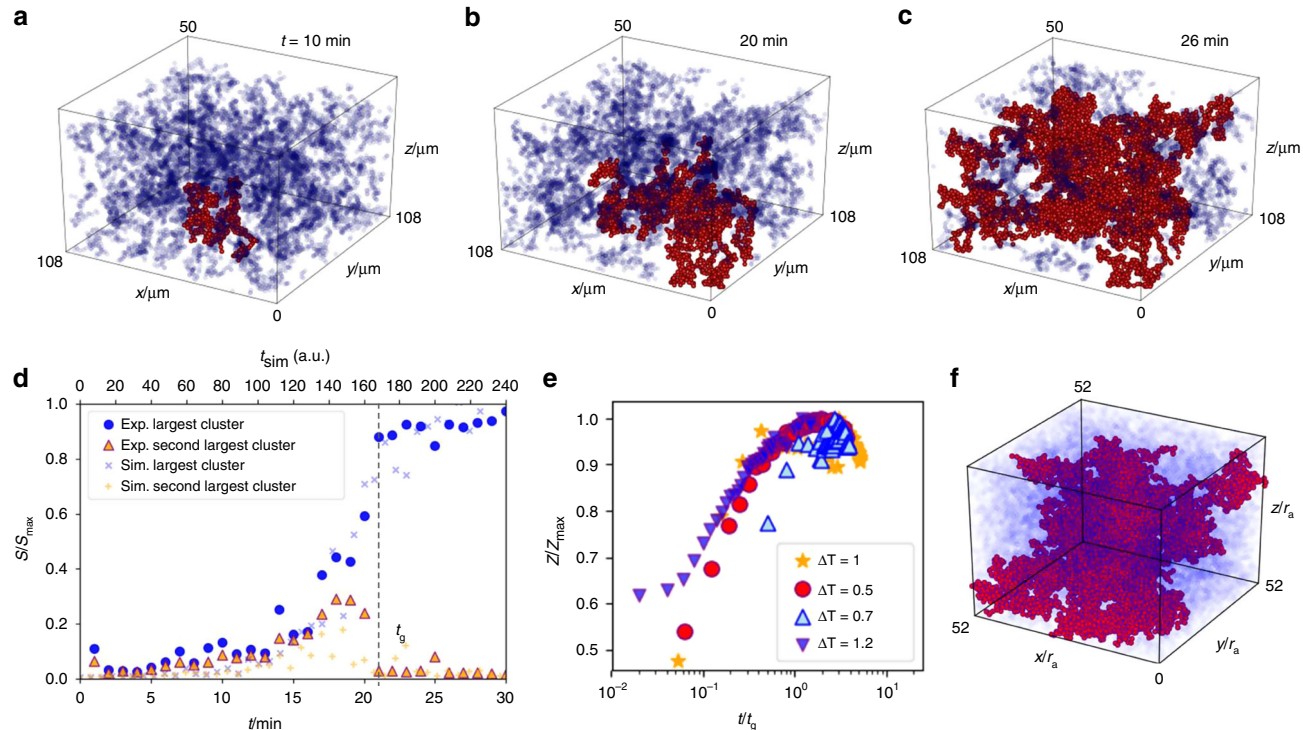

**Fig. 1 Observation of gelation in experiments and simulations. a–c** Experimental observation of aggregating colloidal particles interacting with critical Casimir forces at $\Delta T = 0.5$ K. Largest connected cluster is marked in red. **d** Size of the largest, and second-largest cluster normalized by the maximum cluster size as a function of time. **e** Evolution of the normalized mean coordination number with time in experiments, and in simulations at $\epsilon/k_B T = 4$. **f** aggregated particles in the simulation at $\epsilon/k_B T = 4$ (late stage snapshot).

critical value $z_c = 5.5$, with exponent $\sigma \approx 1.6$. Concomitantly, the length scale of connected particle clusters diverges. We determine the correlation length of connected particles using $\xi^2 = 2\sum_i R_{gi}^2 N_i^2 / \sum_i N_i^2$ where $R_{gi}$ is the radius of gyration for cluster of size $N_i$[40]. This correlation length grows also sharply upon approaching the critical coordination number $z_c$ as shown in Fig. 2b, diverging as $\xi \sim (z_c - z)^{-\nu}$ with $\nu = 0.8$ (inset). This exponent is consistent with three-dimensional percolation results[40]. Similar behavior is observed for all other attractive strength. The same divergence is also observed in the simulations, see Fig. 2c, d, where we compile data for all investigated attractive strengths. All data collapse onto single curves, indicating that the same mechanism applies irrespective of the attractive strengths. We observe divergence of $f_z$ and $\xi$ upon approaching the critical coordination number, again with exponents of $-1.6$ and $-0.8$, respectively, for $f_z$ and $\xi$. The same scaling is observed for simulations based on a square-well potential of similar short range, see Fig. 2e. Furthermore, as shown in the Supplementary Notes 2 and 3, identical scaling is observed over a range of particle volume fractions and attractive strengths, and for a very different short-range attractive experimental system of protein microparticles (Supplementary Figure 3), indicating that the observed divergence is robust and a general property of the gelation.

Percolation occurs only for sufficiently strong attraction; for attractive strength smaller than $\epsilon_c/k_B T = 2.5$, the critical coordination number $z_c$ is no longer reached (green dots for $\epsilon/k_B T = 2$), and the clusters do not span space, consistent with the previously observed cluster phase in depletion systems[2]. We find that upon approaching the critical attraction $\epsilon_c$ from below, the critical coordination number $z_c$ is approached in a power-law fashion (Fig. 2f), giving independent evidence of an underlying critical point.

We thus observe all hallmarks of percolation, while at the same time detailed balance is broken as shown in Fig. 3. Here, we plot association and dissociation rates, measured directly from subsequent simulation snapshots (see Methods and Supplementary Movie 1), as a function of cluster size for different attractive strength. Association rates are clearly larger than dissociation rates, and show a different cluster-size dependence: while the former are roughly size-independent, the latter decrease rapidly with cluster size, being largest for single-particle break off.

**Analytical model**. To interpret the experimental and simulation results within the framework of nonequilibrium statistical mechanics, we study a kinetic master equation for partially reversible aggregation used before in colloidal and protein aggregation[3,41,42] (see Methods). The equation describes changes in the cluster sizes $c_k$ for all clusters $k = 1..N$ owing to dissociation into clusters of $i$ and $j$ particles occurring with rate constant $K_{ij}^-$, and merging of clusters with $i$ and $j$ particles occurring with rate constant $K_{ij}^+$. It is analytically solvable for the physically meaningful case that the dissociation rate constant is non-zero only for single-particle dissociation, and the association rate has the same value for all aggregate sizes. Both assumptions are reasonably well supported by Fig. 3, showing that the attachment rate is fairly independent of the cluster size, and the detachment rate decreases rapidly with cluster size. Physically, the idea is that multiply connected particles belonging to inner cluster shells sit in much deeper energy minima, whereas particles at the surface sit in shallower potential wells, breaking off much more easily, as supported by recent simulations[43]. Under these assumptions, the master kinetic equation simplifies to (see "Methods")

$$\frac{dC}{dt} = C^2 + 2\lambda\frac{1-z}{z}C + 2\lambda\frac{(1-z)^2}{z}N(t) \quad (1)$$

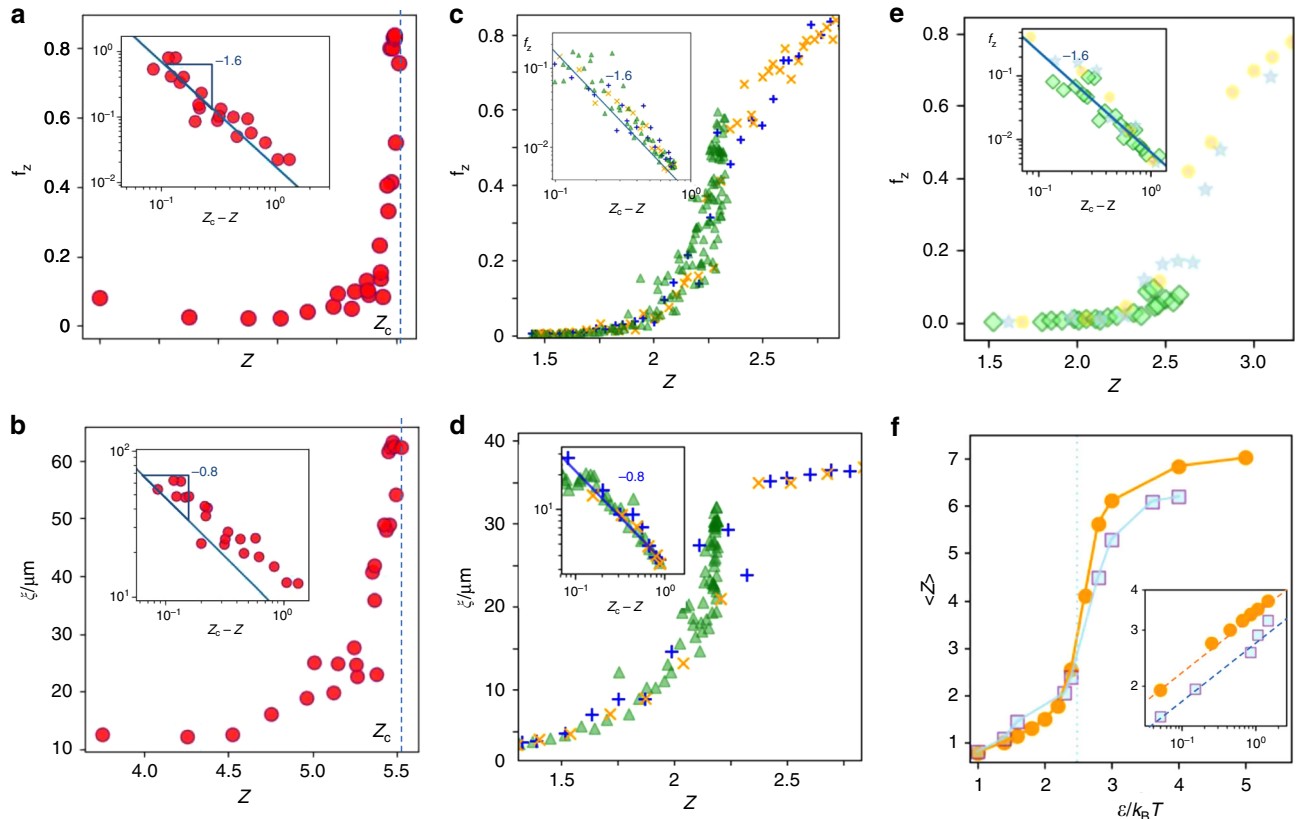

**Fig. 2 Divergence of cluster size at percolation. a** Fraction of particles in the largest cluster, $f_z$ as a function of the mean coordination number, $z$ in experiments at $\Delta T = 0.5$ K. Sharp rise signals percolation. Inset: divergence of $f_z$ upon approaching the critical coordination number $z_c$. **b** Correlation length, $\xi$, as a function of $z$ for experiments at $\Delta T = 0.5$ K. Inset: divergence of $\xi$ upon approaching the critical coordination number, $z_c$. The exponent $-0.8$ is consistent with three-dimensional percolation. **c** Fraction of particles in the largest cluster as a function of mean coordination number for Mie potential simulations at $\epsilon/k_B T = 2$ (green triangles), 3 (blue plus signs), and 4 (yellow crosses). Inset: divergence of $f_z$ upon approaching the critical coordination number $z_c$. **d** Correlation length as a function of $z$ for the Mie potential simulations. Inset: divergence of $\xi$ upon approaching the critical coordination number. **e** Same quantity as in **c** for simulations using a square-well potential. **f** Average steady-state coordination number as a function of attractive strength for Mie (yellow dots) and square-well potential simulations (gray squares). Inset: approach of the critical coordination number with attractive strength approaching the critical attraction $\epsilon_c$ from below. Critical scaling is observed.

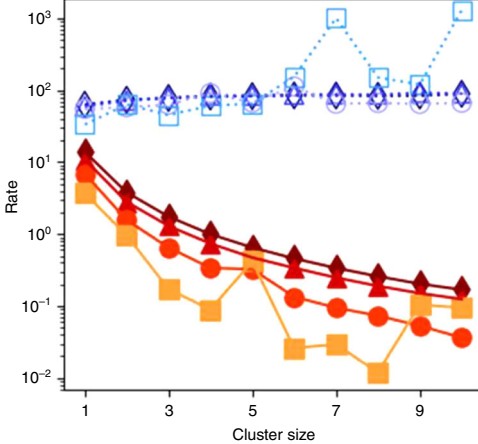

**Fig. 3 Breaking of detailed balance.** Association (open blue symbols, units $\tau^{-1} r_a^{-3}$) and dissociation rates (closed orange to red symbols, units $\tau^{-1}$) of single-bonded particle clusters as a function of cluster size in simulations. Curves from top to bottom indicate increasing $\epsilon/k_B T = 2$ (triangle), 2.2 (diamond), 2.4 (dot), and 2.6 (square), across the gelation transition.

where $C(z, t) = \sum_{j \geq 1}(z^j - 1)c_j(t)$, with $z$ a dummy variable as usually defined in generating functions, $N(t) = \sum_{j \geq 1} c_j(t)$ and we took $K_{ij}^{+} = 2$ for ease of notation and without any loss of generality[44,45]. Here, the parameter $\lambda$ measures the extent of single-particle breakup, and is proportional to $\exp(-V/k_B T)$ with $V$ the depth of the pair attraction well. Clearly, the last condition breaks the detailed balance: there is no linear dependence between aggregation and fragmentation rates for all processes involving $i$ and $j$ both larger than unity, meaning these aggregation processes are de facto irreversible. It follows that any stationary state for which the cluster-mass distribution reaches a steady-state in time is thus a nonequilibrium stationary state. At steady-state, defined by $dC/dt = 0$ for $t \rightarrow \infty$, the second-order algebraic equation is solvable, and differentiating $C$ with respect to $z$ and setting $z = 1$ gives $N$ as a function of $\lambda$. A continuous phase transition at the critical point $\lambda_c = 1$ is found, which separates the sol state with $N = 1 - (2\lambda)^{-1}$ from the gel state (spanning network) with $N = \lambda/2$.

Hence, this model predicts gelation as a continuous (second-order) phase transition, with a cluster-mass distribution that exhibits two distinct power-law exponents, namely $\tau = -3/2$, with an exponential tail, in the sol phase, and $\tau = -5/2$, without the exponential tail, in the gel phase (see "Methods").

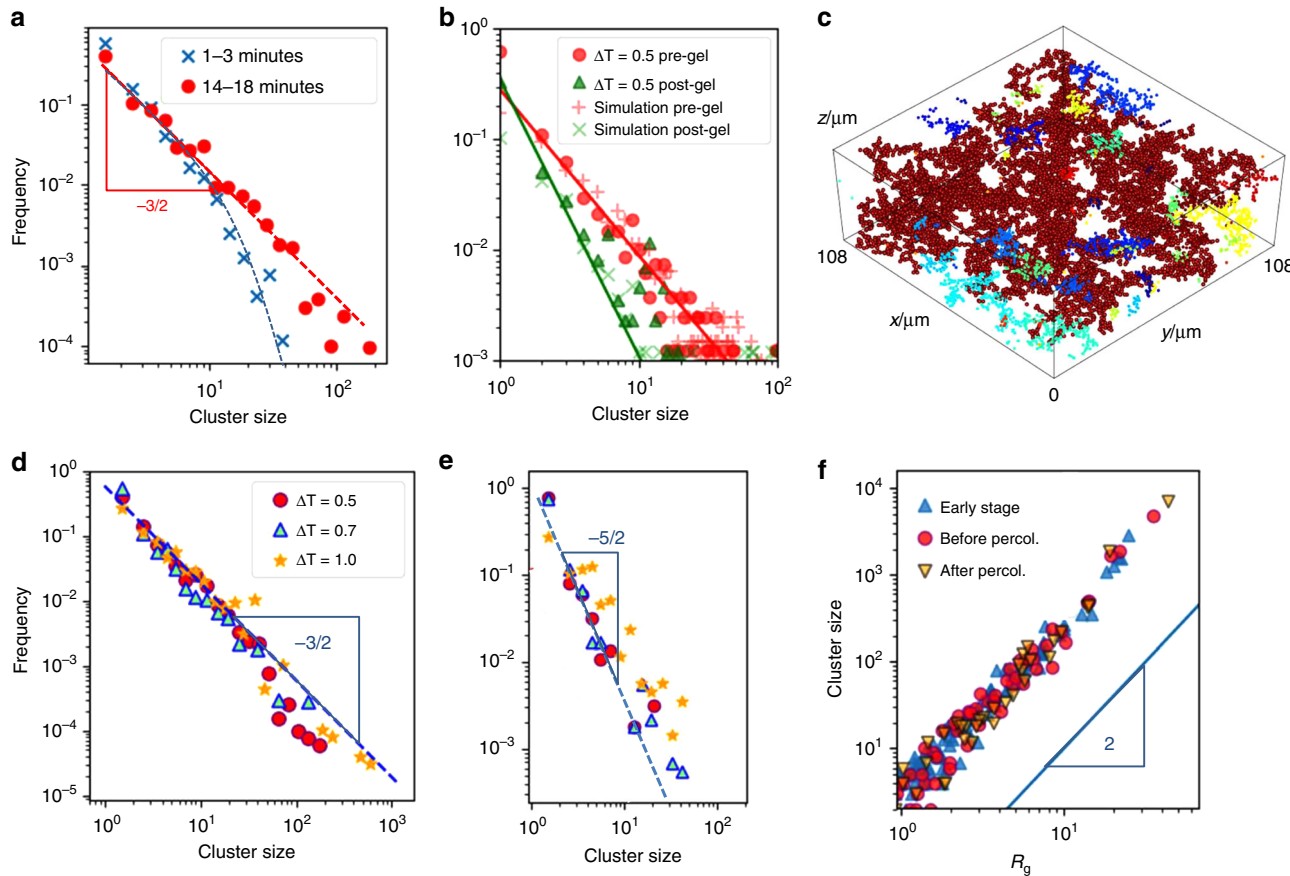

**Fig. 4 Cluster-mass distribution and radius of gyration across the gelation transition. a** Cluster-mass distributions in experiments at $\Delta T = 0.5$ K at early and late stage before percolation. **b** Cluster-mass distribution just before and just after percolation, for experiments (closed symbols) and simulations (crosses). Lines indicate the predicted powers of $-3/2$ and $-5/2$. **c** Reconstruction of the largest cluster (red) and smaller clusters (different colors) in experiment just after gelation. **d, e** Cluster-mass distributions just before **d** and just after gelation **e**, for various attractive strength in experiments. **f** Cluster size versus radius of gyration in experiments at $\Delta T = 0.5K$ for three stages of growth: early stage, before percolation, and after percolation. Each point represents a cluster.

**Verification of model predictions**. To test these predictions, we monitor cluster sizes over time, and determine their distributions just before and just after gelation, as shown in Fig. 4a and b. The initial exponential cutoff grows to larger sizes until a power-law distribution emerges near gelation (Fig. 4a). The data are indeed in good agreement with the predicted exponent $-3/2$ before percolation (see also Supplementary Figure 4). After percolation, a large space-spanning cluster coexists with a dilute population of clusters whose size distribution approaches a power-law with slope close to $-5/2$, as shown in Fig. 4b, where we have taken out the largest cluster and show the distribution of the remaining cluster population. A full reconstruction of the space-spanning cluster coexisting with the smaller clusters is shown in Fig. 4c. The power-law slope of $-5/2$ is again consistent with the analytical prediction. Further confirmation comes from the simulations that show very similar distributions (crosses in Fig. 4b). Furthermore, the cluster-mass distributions are also fairly robust upon variation of the attractive strength as shown by the experimental data in Fig. 4d and e. For all systems reaching percolation, cluster-mass distributions closely follow the predicted power-laws with exponents $-3/2$ and $-5/2$ before and after percolation, respectively. Some deviation is expected as the assumption of single-particle breakup is an approximation. Finally, we can compare the fractal dimension determined experimentally with the value estimated from the hyperscaling relation of standard percolation $\tau = (d/d_f) + 1$,

where $\tau$ is the power-law exponent of the cluster-mass distribution at percolation, assuming that this relation holds also for the nonequilibrium case studied here (recent experimental evidence supporting the validity of hyperscaling relations in colloidal gelation has been shown in ref. [46]). Using $\tau = 5/2$ as predicted by the model and confirmed in both experiments and simulations, we obtain the prediction $d_f = 2$. This is indeed in very good agreement with the experimental data, as shown by plotting the cluster size as a function of radius of gyration in Fig. 4f. Here, we plot data at different stages before and after percolation, and find a robust power-law slope indicating a constant fractal dimension, consistent with $d_f \sim 2$ (solid line); yet, the limited dynamic range does not exclude other possible scenarios (such as e.g., $d_f = 1.8$ for DLCA). We also note that the fractal dimension is expected to increase owing to ageing, leading to more compact structures[32,37]; yet, such aging is not observed on the time scale of our experiment as shown by the constant slope in Fig. 4f. We note that, although the cluster distributions are accurately predicted by the model, properties that require spatial information may not be. As an example, we estimate the critical exponent $\sigma$ from the divergence of the largest (cutoff) size using Eq. (5) in the Methods section. A Taylor expansion of this expression yields a leading term that diverges with power-law of $-2$, different from the exponent $-1.6$ determined in the experiments and simulations. This deviation between our simple mean-field model prediction and

the experimental and simulation results reminds of the deviation of mean-field model predictions of critical exponents in equilibrium critical phenomena.

## Discussion

Our critical Casimir colloidal experiments, simulations, and analytically solvable master kinetic equation description all converge unambiguously to show that the observed gelation of short-range attractive colloids at intermediate densities manifests as a nonequilibrium continuous phase transition with exponents reminiscent of standard percolation in 3$d$. The cluster-mass distributions, predicted by kinetic theory with the assumption of single-particle thermal detachment from clusters, are quantitatively confirmed in both experiments and simulations for the investigated attraction range. Furthermore, application of the hyperscaling relation of equilibrium percolation leads to accurate prediction of the fractal dimension. These results inspire a more general understanding of the fluid-to-solid transition in disordered systems. The yielding of amorphous solids has likewise been identified as a nonequilibrium percolation transition[47,48]. Because the yielding process, which fluidizes an initially solid material, can be regarded as a process opposite to gelation, which solidifies an initially fluid-like sample, it appears that the observed onset of rigidity from a fluid state, and the onset of flow from a solid state are two almost mirror-image manifestations of the same nonequilibrium continuous transition. This general framework comprehends the onset and loss of rigidity as two related, but evolving in opposite directions, nonequilibrium phase transitions. Indeed, the observed robustness of the scaling relations suggests some universality, meaning that the gelation mechanism, at least in this range of attraction and volume fractions, is independent of the precise form of the interparticle interaction potential. This mechanism can hence be used for the tailored self-assembly of a variety of different systems with greatly varying pair interactions, particles, and solvents. It appears that the classically discussed equilibrium percolation[24,25] extends towards a nonequilibrium continuation, governed by very different underlying kinetics (broken detailed balance), of crucial importance in, for example, biological structure formation. Indeed, recent studies on the gelation of random-patchy particles mimicking proteins highlight the direct analogy to adhesive hard spheres and our system[11].

## Methods

**Colloidal suspension**. The colloids are fluorescently labeled copolymer particles made of 2,2,2-trifluoroethyl methacrylate[49] with radius $r_0 = 1$ μm and a poly-dispersity of 5%. The particles are suspended at a volume fraction $\phi \sim 0.12$ in a binary mixture of lutidine and water, with weight fraction of lutidine $c_L = 0.25$. Sugar was added to match the solvent refractive index and density with that of the particles, while only slightly affecting the binary solvent phase diagram. We also added salt (5 mM KCl) to screen the electrostatic repulsion of the charge-stabilized particles, as in previous studies[31,37]. Phase separation of this solvent occurs at $T_c = 31.0$ °C, with a critical composition of $c_c = 0.26$ as determined by systematic investigation of the solvent phase diagram over a range of compositions.

**Experiments**. We use a fast confocal microscope (Zeiss 5 Live) equipped with a ×63 lens with a numerical aperture of 1.4 to image individual colloidal particles in a 108 μm by 108 μm by 60 μm volume. Three-dimensional image stacks with a distance of 0.2 μm between images are acquired every 60 s over a time interval of at least 60 mins to follow the gelation process in three dimensions from the initial cluster formation to gelation and beyond. During this process, the temperature is kept strictly constant by using a specially designed water heating setup that controls the temperature of both the sample and the coupled oil-immersion objective with a stability of ~0.01 °C. Particle positions are determined from the three-dimensional image stacks with an iterative tracking algorithm to optimize feature finding and particle locating accuracy[50]. The resulting particle positions have an accuracy of ~20 nm in the horizontal and ~50 nm in the vertical direction. To show this, we used several layers of particles stuck to a cover slip, which we imaged and located repeatedly to determine histograms of particle positions. From this, we determine the positional variances $\sigma_x = 15$ nm, $\sigma_y = 20$ nm, and $\sigma_z = 40$ nm. From the

determined three-dimensional particle positions, bonded particles are identified as those separated by $< d_0 = 2.6\,r$, corresponding to the first minimum of the pair correlation function. We subsequently group bonded particles into connected clusters using a clustering algorithm based on a threshold distance of $d_c = 3.5\,r$.

**Simulations**. Molecular dynamics simulations are used to model the trajectories of particles with radii $r_a/r_b = 1:1.1$ mixed equally at volume fraction $\phi = 0.12$, interacting through a Mie potential, with attractive range matching that of the experiments and attractive strength given by prefactor $\epsilon$. The potential acts between all particle pairs within a cutoff range $r_c = 1.5r_a$. The kinetic state of our system, liquid or gel, is determined by the dimensionless control parameter $\epsilon/k_B T$, where $k_B T$ is the thermal energy. The time unit is $t_s = \sqrt{mr_a^2/\epsilon}$, with $m$ the mass of a particle with radius $r_a$. Particle trajectories follow the Langevin equation[51], with coefficient of friction $1/\zeta$ (we set $\zeta = t_s$) and random forces $f_B(t)$ satisfying $\langle f_B(t) f_B(t') \rangle = 2mk_B T \delta(t - t')/\zeta$. Lennard-Jones units are used throughout to maintain generality, and we use $dt = 0.0025\,t_s$ as the numerical time step. Simulations are performed in a cubic box with periodic boundary conditions containing $N = 32,768$ particles. Systems are equilibrated in the liquid state with $\epsilon/k_B T = 1$ before switching to larger values and following the subsequent cluster formation. Contacting particles are those within the inflection point of the potential, which in this case occurs at $(31/14)^{0.1} r_a$. This allows us to identify clusters, and follow the evolution of their size distributions across the gelation transition for a range of attractive strengths.

We also perform simulations with an approximate square-well potential. We adopt the "continuous square-well" model described in ref. [52], writing the potential as

$$U_{csw}(r) = \frac{1}{2}\,\epsilon\left(\left(\frac{1}{r}\right)^n + \frac{1 - e^{-m(r-1)(r-r_{sw})}}{1 + e^{-m(r-1)(r-r_{sw})}} - 1\right), \qquad (2)$$

using a binary form for the width of the well $r_{sw}$ (potential range) to match our particle size ratio for the Mie potential. The dimensionless well steepnesses $m$ and $n$ are set as 7000 and 700, respectively, leading to a second virial coefficient (defined following ref. [53]) that matches that of the Mie potential at $\epsilon/k_B T = 3$.

**Calculation of association and dissociation rates**. To calculate association and dissociation rates, we first define directly contacting particles as those whose centers lie within the inflection point of the potential (where $\frac{\partial^2 U}{\partial r^2} = 0$). Based on these criteria, we define a particle as belonging to a cluster if there exists a continuous series of direct contacts between that particle and all other particles in the cluster. Outputting the particle coordinates with very fine time resolution then allows us to monitor the temporal evolution of cluster sizes throughout the system as successive dissociation and association events occur, and thus to compute the rate constants $K_{ij}^{+/-}$ in the kinetic master equation, see below. Here, $K_{ij}^+$ means the association rate of clusters that have, respectively, $i$ and $j$ particles, whereas $K_{ij}^-$ indicates the split-up or dissociation rate of a larger cluster into clusters of $i$ and $j$ particles. We determine the rate of dissociation events involving clusters of size 4, for example, by averaging dissociation rate $K_{4j}^-$ over $j$. As a result, we find that the rate of detachment of whole clusters is considerably smaller than that of events where a single particle detaches from a cluster, whereas the rates of the corresponding association events are comparable.

**Cluster growth model**. We start with the master kinetic equation for the time-evolution of the cluster population $c_k$, i.e., the number of clusters with $k$ particles per unit volume:

$$\frac{dc_k}{dt} = \frac{1}{2}\sum_{i+j=k} K_{ij}^+ c_i c_j - c_k \sum_{j\geq 1} K_{kj}^+ c_j + \sum_{j\geq 1} K_{kj}^- c_{j+k} - c_k \sum_{i+j=k} K_{ij}^-. \qquad (3)$$

In this master equation, the first term on the right hand side represents the creation of clusters with $k$ units owing to aggregation of one cluster with $i$ units with another with $j$ units (where $i + j = k$); the second term represents the "annihilation" of clusters with $k$ units owing to aggregation of a cluster with $k$ units with a cluster of any other size in the system; the third term represents creation of a cluster with $k$ units owing to the breakage of a larger cluster which splits into a cluster with $k$ units and another of $j$ units, where $j$ can take any value; the fourth term represents annihilation of a cluster with $k$ units owing to fragmentation into two fragments $i$ and $j$, subjected to mass balance. This equation can only be solved numerically. However, in the case of only single-particle detachment, and cluster-size independent attachment, an analytical solution is available. Upon introducing the generating function (a procedure similar to a discrete Laplace transform) $C(z, t) = \sum_{j\geq 1}(z^j - 1)c_j(t)$, where $z$ is a dummy variable as usually defined in generating functions, the system of ordinary differential equations is reduced to the Riccati equation (Eq. (1) in the manuscript), yielding a continuous phase separation for $\lambda_c = 1$. By expanding $C(z)$ in powers of $z$ one obtains the cluster-mass distribution in the sol and the gel phase. In the pre-critical sol phase, the

power-law is accompanied by an exponential cutoff[44,45],

$$c_k(t \to \infty) \sim k^{-3/2} e^{-k/k_c}. \tag{4}$$

The presence of the exponential cutoff implies that all clusters are finite in size. However, the cutoff size $k_c$ diverges at $\lambda \to 1^+$, according to [44,45]

$$k_c = \{2\log(\lambda/\lambda_c) - \log[2(\lambda/\lambda_c) - 1]\}^{-1}. \tag{5}$$

In the gel phase $\lambda \geq 1$, the steady-state cluster-mass distribution is

$$c_k(t \to \infty) \sim k^{-5/2}, \tag{6}$$

now without an exponential tail, which signals the existence of a giant system-spanning cluster via the divergence of the first-moment of the distribution. Hence, this model predicts gelation as a continuous (second-order) phase transition, with a cluster-mass distribution that exhibits two distinct power-law exponents, namely $\tau = -3/2$, with an exponential tail, in the sol phase, and $\tau = -5/2$, without the exponential tail, in the gel phase.

## Data availability

The data sets generated during and/or analyzed during the current study are available from the corresponding author on reasonable request.

## Code availability

The codes of the computer simulations are available from the corresponding author upon request.

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

## Acknowledgements

The authors are grateful to Francesco Sciortino for useful comments, and to Carlijn van Balen and Erik van der Linden for preparing the protein microparticle system. This work is funded by an industrial partnership program, subsidized by the Netherlands Organization for scientific research (NWO). P.S. acknowledges support by a Vici Fellowship from NWO. C.N. acknowledges financial support from the Maudslay-Butler Research Fellowship at Pembroke College, Cambridge, and latterly from the Royal Academy of Engineering under the Research Fellowship scheme. A.Z. gratefully acknowledges financial support from the US Army Research Office through contract nr. W911NF-19-2-0055.

## Author contributions

J.C.R., A.Z., and P.S. conceived the study. J.C.R. performed the experiments and analyzed the data. C.N. performed the simulations. J.C.R. and P.S. wrote the paper except the modeling part, written by A.Z., and the simulation part, written by C.N. S.S. advised on the study and manuscript. All authors discussed the data and reviewed the manuscript.

## Competing interests

The authors declare no competing interests.
