## [Peer Review File · Nature Communications]

Reviewers' comments:

Reviewer #1 (Remarks to the Author):

Review of "Nonequilibrium continuous phase transition in colloidal gelation with short-range attraction" by Rouwhorst et al.

This paper presents a combined, experimental, numerical and analytical study of gelation in attractive colloids. The authors claim that their results provide clear evidence of a non-equilibrium percolation transition. This is a strong claim, and if justified would certainly be a story worth publishing in Nature Communications. The current manuscript, however, does not meet this criterion, in my opinion. I will point out the scientific claims that I find unconvincing below but the paper also needs to be rewritten to make it less confusing for the reader.

1. My main issue is with the authors claiming to provide strong evidence for a percolation transition, which they conflate with the results of a clustering model that has no *spatial* information in it. The kinetic master equation model can only address cluster sizes and distributions: Eq. 3 in the paper has no spatial information in it. References 35 and 36 do not address spatial aspects such as the fractal dimension of the clusters that arise from aggregation. There are two distinct aspects of gelation: the cluster distribution and the spatial properties of the clusters.
2. The authors first present "evidence" for a percolation transition (Fig. 2), then they present data for cluster size distribution etc. in Fig. 4. These are the properties that can be predicted by the master equation. The master equation does provide strong evidence for a non-equilibrium phase transition that, as shown in Refs 35 and 36, strongly resemble Bose Condensation. There should be an "infinite" aggregate in addition to the distribution of masses in the "gel" phase. Fig. 4 (b) should show this. In the pre-gel phase there is an exponential distribution that should approach a power law, which they do show. As far as I know, the model that the authors keep referring to [Refs. 35 and 36] show that exactly at the gelation point, the power law is 5/2 but beyond this, there is also an infinite aggregate. The authors should clarify if this is indeed what they see.
3. Coming to the spatial properties, the authors claim that the fractal dimension that they observe ($d_f = 2$) is in agreement with predictions. Predictions from what type of percolation model? The only analytical model they present in the paper cannot predict fractal properties of the cluster.
4. The percolation properties are discussed in the context of the average coordination number in the largest cluster, and the correlation length *a la* Ref. 32. The claim that the exponent in Fig. 2b is 0.8 is not very convincing. Then the authors seem to suggest, without explicitly saying so, that f_z diverges with the exponent, σ , characterizing how the maximum cluster size (the cutoff size in the cluster size distribution) in a percolation problem diverges. However, following their analysis of Eq. 3, the master equation, the cutoff size k_c does not diverge with an exponent of $\sigma = 1.6$ (authors claim $d_f = 2$ and $\nu = 0.8$). The authors are using hyperscaling relations appropriate for a classic percolation problem but the "aggregation" problem that they are analyzing (refs. 35 and 36), as far as I know does not map to standard percolation. In fact, I do not know what the exponent " ν " is if I consider a 3D version of the kinetic model that has no spatial information in it.

The above are my main issues with the scientific content of the paper. The authors need to clearly distinguish what information they can obtain from their analytic model, which has no spatial information and where they are obtaining the “predictions” about the spatial structures of the clusters that they compare their numerical simulations to. The critical exponent that they want to associate with the coordination number of the largest cluster needs to be explained.

In addition to these technical issues, I find the writing of the paper to be not very reader friendly. The abstract talks about yielding as a percolation transition, and the authors seem to want to show that gelation is also. The stronger argument in the paper, if they don't start from this yielding perspective, is that gelation is a non-equilibrium phase transition. The “second-order” nature of this is closer to that of a “condensation” transition that they can show by showing the appearance of an “infinite” aggregate in the gel phase in addition to the power law with $5/2$ exponent (cut off by system size) as, for example, in Ref. 35. They can provide clear, clean evidence of this from the data and their model. The coordination number of particles in the cluster etc are interesting but I don't see a clear theoretical basis nor convincing arguments from the data.

There are also grammatical issues that the authors should fix.

Reviewer #2 (Remarks to the Author):

This manuscript studies the formation of colloidal gels using confocal microscopy and supporting computational studies. The significance of this work is to reveal a "nonequilibrium continuous phase transition" through the development of a population balance model of the growing particle clusters (similar, perhaps, to how Smoluchowski would have approached the problem.) The work is definitely interesting and the empirical results robust. However, there are a number of questions and concerns that come to mind. I list several below. Mainly, however, the work is fairly broad in describing the application of the results and conclusions to a more general class of problems, but the experimental range and unusual mechanism of attraction (critical Casimir interactions mediated by a near-critical mixture) makes me hesitant to accept this broad applicability. For one, the range of attractive strengths studied and the volume fraction (just one) is a fair deal more limited than studies such as Ramakrishnan and co-workers, who go to much stronger attractions (using depletion) and a wider range of volume fractions (see Ramakrishnan, Y.-L. Chen, K. S. Schweizer, C. F. Zukoski, *Phys. Rev. E* 70, 40401, 2004.) Interestingly, the gel point coincides at the volume fraction studied here with Cheiw and Glandt's classic dynamic percolation line (for attractive hard spheres) but in depletion systems, at least, the gel line follows close to, and perhaps a little above, the metastable L-L binodal (again, reported by Ramakrishnan et al. as well as by Lu et al.) Even Eberle and co-workers report a gel line that increases above this binodal, yet fall short of the percolation line, but all of the depletion studies that I'm aware of are much lower. Perhaps this is due to the nature of particle attraction in the solvent-mediated silica suspensions that Eberle et al. use.

The other hesitation I have (again, with great interest in reading the work) is the idea that the second-order phase transition analogy of percolation phenomenon is in any way new. This is well known in the chemical gel literature and the subject of many studies, including pioneering work using the critical scaling of structural and rheological properties near the gel point (see the classic works of Winter and Chambon and of Adolf and Martin, for instance - e.g. J. E. Martin, D. Adolf, J. P. Wilcoxon. *Phys. Rev. Lett.* 61, 2620–2623, 1988.)

Below are other questions and comments.

The population balance model employed in this work is interesting and reflects dynamics that are dominated (at least in breakage) by lower coordination number particles on the surface the cluster structures. Zia et al. provide similar insights in their simulations, which studied the long-time coarsening (see R. N. Zia, B. J. Landrum, W. B. Russel, *J. Rheol.* 58, 1121–1157, 2014.) This seems to support the author's interpretation, especially in the rather low attractive strengths studied in both works.

The attractive strength due to the critical Casimir interaction is mapped onto the adhesive hard sphere phase diagram. This is appropriate and discussed by Noro and Frenkel (a reference worth citing) in their work investigating the corresponding states of attractive particles. Note that the L-L binodal (plotted in Supplemental Figure 1) is known from this work to become metastable with respect to the S-L binodal (not shown) when the range of attraction is shorter than $0.14a$. This is very close to the range of attraction reported in this work for the critical Casimir interaction, which is also longer range than other model attractive systems studied by Lu et al., Ramakrishnan et al., and others who have used the depletion interaction to induce attraction.

The gel boundary is interesting to note as well. Varga and Swan report that the the gel boundary for

sticky spheres is sensitive to hydrodynamic interactions. At a volume fraction 0.12, the expected gel boundary from their work with far-field hydrodynamics is close to the experimental boundary reported by Campbell et al. (see Varga and Swan, 2016 for a discussion). In the absence of hydrodynamic interactions, as would be the case for the methods used here, the boundary is shifted to a higher attractive strength. I cannot understand this discrepancy.

Along similar lines, how, then, does the critical transition of the lutidine mixture affect hydrodynamic interactions in the suspension? Are the near-field and / or far-field hydrodynamics altered? (I would assume so, given that the viscosity changes in the vicinity of the mixture phase separation. How does this affect the formation and breakage times of the colloidal "bonds"?)

The critical value of the coordination number is reported as $z_c = 5.5$ for the experiments. It doesn't look like this is the same value in the simulations, but a value is not given. Instead, z_c for the simulations appears to be much lower. Why?

The results don't really create a framework that unify structural arrest and yielding. This is highlighted late in the conclusions without any sort of empirical support by the current work. This would seem to imply, based on the assumptions of the model presented here, that yielding consists of some sort of surface erosion of the cluster, but perhaps I am misunderstanding their idea.

Minor question: How can the entirety of the growing (largest) cluster be completely observed within the imaging volume?

Reviewer 1

This paper presents a combined, experimental, numerical and analytical study of gelation in attractive colloids. The authors claim that their results provide clear evidence of a nonequilibrium percolation transition. This is a strong claim, and if justified would certainly be a story worth publishing in Nature Communications. The current manuscript, however, does not meet this criterion, in my opinion. I will point out the scientific claims that I find unconvincing below but the paper also needs to be rewritten to make it less confusing for the reader.

We thank the reviewer for their strong interest in our paper, and their reviewing effort and valuable comments. We believe that some crucial comments (lack of spatial information, lack of infinite cluster) are indeed based on misunderstandings due to lack of clarity of the original manuscript. We have significantly improved the writing and presentation of the manuscript, and provide additional data and graphs to set these points straight. We furthermore included significant new data (simulations over a range of volume fractions and a very different short-range attractive experimental system), see revised SI and Fig. A1 below. These additional results provide strong further support of our claim of a nonequilibrium percolation transition and demonstrate its generality.

1. My main issue is with the authors claiming to provide strong evidence for a percolation transition, which they conflate with the results of a clustering model that has no spatial information in it. The kinetic master equation model can only address cluster sizes and distributions: Eq. 3 in the paper has no spatial information in it. References 35 and 36 do not address spatial aspects such as the fractal dimension of the clusters that arise from aggregation. There are two distinct aspects of gelation: the cluster distribution and the spatial properties of the clusters.

Spatial information is provided by both experiments and simulations, latter performed for two different potential types. We agree that the analytic model itself is a simple mean-field model based on kinetic equations, and as such has no structural information in it (although we note that it does allow an estimate of the fractal dimension from the predicted cluster mass distribution and inferred hyperscaling relation). The simple model served as an initial guidance for us to design and interpret the experimental and numerical data. This is why we combine it with both experiments and simulations, which do offer plenty of structural information. We stress that as shown below, the crucial mean-field estimates of the model (occurrence of a critical point characterized by infinite cluster as a function of attraction strength, cluster-mass distributions before/after percolation, and fractal dimension) are consistent with the experimental data and simulations. We regard this consistency of three independent methods (analytic model, experiments, simulations), together with the verification of validity of the hyperscaling relations as a strong evidence in favour of our claim of a nonequilibrium phase transition. We apologize if the presentation has not been clear enough e.g. in terms of lack of an infinite cluster (comment 2 below) and consistency of model/data, and we have substantially improved the presentation to eliminate possible misunderstandings.

2. The authors first present “evidence” for a percolation transition (Fig. 2), then they present data for cluster size distribution etc. in Fig. 4. These are the properties that can be predicted by the master equation. The master equation does provide strong evidence for a non-equilibrium phase transition that, as shown in Refs 35 and 36, strongly resemble Bose Condensation. There should be an “infinite” aggregate in addition to the distribution of masses in the “gel” phase. Fig. 4 (b) should show this. In the pre-gel phase there is an exponential distribution that should approach a power law, which they do show. As far as I know, the model that the authors keep referring to [Refs. 35 and 36] show that exactly at the gelation point, the power law is 5/2 but beyond this, there is also an infinite aggregate. The authors should clarify if this is indeed what they see.

Yes, indeed this is what we see! At gelation, we observe an infinite aggregate in addition to the cluster mass distribution with power 5/2. We have taken this infinite aggregate out following the usual convention, and show only the remaining cluster distribution in Fig. 4b; we apologize if this has caused confusion, and we have clarified this in the revised version. To be most clear, we have also added a 3D reconstruction showing the percolating cluster in coexistence with smaller clusters (new Fig. 4c). The revised text reads:

“After percolation, a large space-spanning cluster coexists with a dilute population of clusters whose size distribution approaches a power-law with slope close to -5/2, as shown in Fig 4b, where we have taken out the largest cluster and show the distribution of the remaining cluster population. A full reconstruction of the space-spanning cluster coexisting with the smaller clusters is shown in Fig. 4c.”

We thank the referee for emphasizing this important point, and hope that we have herewith resolved it.

3. Coming to the spatial properties, the authors claim that the fractal dimension that they observe ($d_f = 2$) is in agreement with predictions. Predictions from what type of percolation model? The only analytical model they present in the paper cannot predict fractal properties of the cluster.

Indeed, the model has no structural information in it; we thus inferred the fractal dimension from the predicted cluster mass distribution and the hyperscaling relation of percolation, $\tau = (d/d_f) + 1$. Here, $\tau = 5/2$ is the predicted power-law exponent of the cluster-mass distribution

at percolation, and $d=3$ is the spatial dimension, resulting in $d_f=2$. Our argumentation is as follows: if the gelation is indeed related to a kind of percolation transition, and the hyperscaling relation holds for our non-equilibrium case, this would predict $d_f=2$, which is indeed what we find in the experiments and simulations. We apologize if this has not been clear; we have rewritten this paragraph to clearly state: “Finally, we can compare the fractal dimension determined experimentally with the value estimated from the hyperscaling relation of percolation $\tau = (d/d_f) + 1$, where τ is the power-law exponent of the cluster-mass distribution at percolation. Using $\tau = 5/2$ as predicted by the model and confirmed in both experiments and simulations yields the prediction $d_f = 2$.”

In general, we have toned down claims of the model being “predictive”, as clearly there is no analytic theory, to date, which can provide a detailed description (including spatial information) of kinetic nonequilibrium percolation processes. We believe that the simple mean-field model proposed here can capture and explain some aspects of the weak gelation process and has proved useful in directing the experimental/simulation efforts. Historically, also the Landau theory of phase transitions provided only a mean-field estimate of critical exponents without structural information, which only later were refined by the theory of critical phenomena. Nevertheless, it played a crucial role in advancing our understand of equilibrium phase transition.

4. The percolation properties are discussed in the context of the average coordination number in the largest cluster, and the correlation length a la Ref. 32. The claim that the exponent in Fig. 2b is 0.8 is not very convincing. Then the authors seem to suggest, without explicitly saying so, that f_z diverges with the exponent, σ , characterizing how the maximum cluster size (the cutoff size in the cluster size distribution) in a percolation problem diverges. However, following their analysis of Eq. 3, the master equation, the cutoff size k_c does not diverge with an exponent of $\sigma=1.6$ (authors claim $d_f = 2$ and $\nu = 0.8$). The authors are using hyperscaling relations appropriate for a classic percolation problem but the “aggregation” problem that they are analysing (refs. 35 and 36), as far as I know does not map to standard percolation. In fact, I do not know what the exponent “ ν ” is if I consider a 3D version of the kinetic model that has no spatial information in it.

As the experimental data leaves some ambiguity in the scaling exponent (0.8) of the correlation length, we have added the simulation data of the correlation length (new Fig. 2d), which shows a clear scaling with slope -0.8 for all potentials. We believe that together (experiments, and simulations with two different potentials), these data provide strong evidence for the described scaling behaviour. Concerning the hyperscaling relations, we indeed apply them outside their usual equilibrium percolation context, in a new nonequilibrium (aggregation) regime, which we believe is a novelty of our paper. Our argumentation is as follows: because we observe, both in experiments and simulations, the divergence of cluster correlation lengths in agreement with 3D percolation models, we apply the hyperscaling relation (together with the measured and predicted cluster mass distribution exponent 5/2), and obtain a fractal dimension, which is indeed in agreement with measurements. We thus establish an independent “closure”, which we regard as additional evidence for our claim, i.e. the occurrence of a percolation transition in the nonequilibrium regime. The fact that the model predicts the right cluster mass distributions, without spatial information in it, is a possible hint that these exponents are general, and do not rely on specific structural details.

In the absence of a theory of nonequilibrium percolation phenomena, one cannot explain, on a fundamental level, why the hyperscaling relation seems to work in combination with the simple mean-field kinetic model that we used. We believe, however, that these results can guide and steer further theoretical efforts in future work aimed at understanding nonequilibrium percolation.

Concerning the predicted exponent σ , the referee makes a good remark, which we haven't thought about before. Indeed, the simple mean-field kinetic model gives a divergence of largest (cut off) size given by Eq. (5) in the Methods section. A Taylor expansion of this expression yields a leading term that diverges with power-law of -2 , which deviates from the exponent -1.6 of the experiments and simulations. However, this is not too surprising for a simple mean-field model that has no structural information in it. As the referee remarks, there are two distinct aspects of gelation: the cluster distribution and the spatial properties of the clusters. Former can be, and is accurately predicted by the model, while absence of the latter does not allow to accurately predict the divergence of the largest cluster size. In fact, the deviation reminds of that of critical exponents of mean field models of equilibrium phase transitions. We thank the referee for pointing this out clearly, and have added a corresponding remark in the revised manuscript:

“We note that, while the cluster distributions are thus accurately predicted, properties that require spatial information may not. As an example, we estimate the exponent σ from the divergence of the largest (cut off) size using eq.5 in the Methods section. A Taylor expansion of this expression yields a leading term that diverges with power-law of -2 , different from the exponent -1.6 determined in the experiments and simulations. This deviation between our simple mean-field model prediction and the experimental and simulation results reminds of the deviation of mean-field model predictions of critical exponents in equilibrium critical phenomena.”

Reviewer #2 (Remarks to the Author):

This manuscript studies the formation of colloidal gels using confocal microscopy and supporting computational studies. The significance of this work is to reveal a "nonequilibrium continuous phase transition" through the development of a population balance model of the growing particle clusters (similar, perhaps, to how Smoluchowski would have approached the problem.) The work is definitely interesting and the empirical results robust. However, there are a number of questions and concerns that come to mind. I list several below. Mainly, however, the work is fairly broad in describing the application of the results and conclusions to a more general class of problems, but the experimental range and unusual mechanism of attraction (critical Casimir interactions mediated by a near-critical mixture) makes me hesitant to accept this broad applicability. For one, the range of attractive strengths studied and the volume fraction (just one) is a fair deal more limited than studies such as Ramakrishnan and co-workers, who go to much strong attractions (using depletion) and a wider range of volume fractions (see Ramakrishnan, Y.-L. Chen, K. S. Schweizer, C. F. Zukoski, Phys. Rev. E, 70, 40401, 2004.) Interestingly, the gel point coincides at the volume fraction studied here with Cheiw and Glandt's classic dynamic percolation line (for attractive hard spheres) but in depletion systems, at least, the gel line follows close to, and perhaps a little above, the metastable L-L binodal (again, reported by Ramakrishnan et al. as well as by Lu et

al.) Even Eberle and co-workers report a gel line that increases above this binodal, yet fall short of the percolation line, but all of the depletion studies that I'm aware of are much lower. Perhaps this is due to the nature of particle attraction in the solvent-mediated silica suspensions that Eberle et al. use.

We thank the reviewer for their interest in our work, and acknowledgement of the robustness of our results. To further show the generality of our results, we have included a broader range of volume fractions, and an additional very different experimental system: protein microparticles. All additional systems together with those of the original manuscript converge to the same conclusion, as shown in Fig. A1 below. Altogether, the manuscript now reports two different experimental systems, two simulation systems (with different pair potential) over a range of volume fractions, and a theoretical model that all converge to support our original claim of a nonequilibrium continuous phase transition. We believe that the use and agreement of such diverse range of methods is unusual for a single article, and hope that the referee can agree to our conclusion on a general underlying mechanism (at least within the regime of short-range attractive systems studied). Concerning the critical Casimir effect, while we agree that this is a more recent “addition” to controlling colloidal interactions, it is by now (since a few years) well established, and has been widely shown to provide an effective interaction controlling colloidal phase behaviour (Hertlein *et al. Nature* (2008), Guo *et al., Phys. Rev. Lett.* (2008), Mohry *et al., J. Chem. Phys.* 136, 224902 (2012) and 224903 (2012); Dang *et al. J. Chem. Phys.* (2013), Nguyen *et al. Nature Comm.* (2013), Mohry *et al. Soft Matter* (2014), Stuij *et al. Soft Matter* (2017), and many others as reviewed in Nguyen *et al. J. Phys.: Cond. Matt.* 28, 043001 (2016) and Maciolek and Dietrich *Rev. Mod. Phys.* 90, 045001 (2018)). Its advantage is the use of an effective solvent-mediated interaction without the need for an extra component (e.g. depletant). We have verified this and quantified the interaction by elaborate comparison of theoretical prediction and experimental measurements of colloidal interactions as detailed in Stuij *et al., Soft Matter* **13**, 5233 (2017). Because of this effective nature of the interaction and the larger separation of scales, we believe that this is actually an ideal system to study aggregation and gelation phenomena in their most general form, unlike e.g. the depletion interaction that is based on an underlying colloid-polymer phase separation. We have previously studied colloidal aggregation by critical Casimir forces in microgravity (on the International Space Station), and observed consistent behaviour in agreement with models and earlier work (Veen *et al. Phys. Rev. Lett.* **109**, 248302 (2012), Potenza *et al. Europhys. Lett.* **106**, 68005 (2014), Potenza *et al. Europhys. Lett.* **124**, 28002 (2018)). We believe that all these published works on critical Casimir interactions no longer make this an unusual interaction, but rather - seeing the effective interaction and large separation of scales – a very suitable interaction for general insight into phase transition and dynamic arrest phenomena. To better highlight this point we have included more references and extended the sentence: “**Previous studies have revealed equilibrium phase transitions from gas to liquid and liquid to solid at low attraction [Guo2008,Nguyen2013,Dang2013,Nguyen2018], as well as colloidal aggregation at higher attraction, which was investigated in microgravity [Veen12,Potenza2014,Potenza2018]. To study gelation, we induce sufficiently strong attractive strength between the particles by ...**”

Finally, by significantly broadening the range of volume fractions, we now provide a full gelation diagram in the supplementary information, which shows a decrease of the gelation line far below the spinodal towards low volume fraction, qualitatively consistent with the work of

Fig. A1 Gelation at various volume fractions and in different systems. (a) Fraction of particles in the largest cluster as a function of the average mean coordination number, in simulations for particle volume fractions $\phi = 0.06, 0.12$ and 0.16 , and different attractive strength, and in an experimental system of whey protein isolate (WPI) microparticles (see legend). (b) Same data as a function of normalized coordination number z/z_c . Inset: Divergence of f_z upon approaching the critical coordination number z_c . All systems show the same divergence with slope 1.6, consistent with percolation theory.

Eberle *et al.* All additional volume fractions (and the very different experimental system, protein microparticles), show the same scaling upon approaching the gelation point, confirming our previous interpretation, and lending further support to the generality of our results. These additional data sets are included in the supplementary information.

The other hesitation I have (again, with great interest in reading the work) is the idea that the second-order phase transition analogy of percolation phenomenon is in any way new. This is well known in the the chemical gel literature and the subject of many studies, including pioneering work using the critical scaling of structural and rheological properties near the gel point (see the classic works of Winter and Chambon and of Adolf and Martin, for instance - e.g. J. E. Martin, D. Adolf, J. P. Wilcoxon. Phys. Rev. Lett. 61, 2620–2623, 1988.)

We are well aware of the mentioned works, referring to the well-known Flory-Stockmayer gelation models, but they all refer to *equilibrium* percolation. Thus, all these works relate to equilibrium transitions, where equilibrium statistical mechanics based on detailed balance applies. Our work, however, is about a non-equilibrium transition, as proven in the manuscript (with both simulations and experiments) by the breaking of detailed balance, which is the signature of a nonequilibrium phase transition that is conceptually very different. We hence believe that the crucial novelty of our work is the existence of a *non-equilibrium* critical point (nonequilibrium percolation transition), which is very different and has never been shown before, as acknowledged by Reviewer 1. We believe that the significance of this nonequilibrium percolation transition becomes even more evident in a broader context of flow-arrest transitions, as the yielding transition (arguably the “mirror transition” of aggregation/gelation) has been recently shown to be governed by a related nonequilibrium percolation transition (Shrivastav *et al.* *Phys. Rev. E* **94**, 042605 (2016), Ghosh *et al.* *PRL* **118**, 148001 (2017)) see our more detailed explanation below.

Below are other questions and comments.

The population balance model employed in this work is interesting and reflects dynamics that are dominated (at least in breakage) by lower coordination number particles on the surface of the cluster structures. Zia et al. provide similar insights in their simulations, which studied the long-time coarsening (see R. N. Zia, B. J. Landrum, W. B. Russel, J. Rheol. 58, 1121–1157, 2014.) This seems to support the author's interpretation, especially in the rather low attractive strengths studied in both works.

Thank you for this nice confirmation. We have added a remark with the reference in the text. We modified the corresponding sentence to: “Physically, the idea is that multiply connected particles belonging to inner cluster shells sit in much deeper energy minima, while particles at the surface sit in shallower potential wells, breaking off much more easily, **as supported by recent simulations [Russel2014].**”

The attractive strength due to the critical Casimir interaction is mapped onto the adhesive hard sphere phase diagram. This is appropriate and discussed by Noro and Frenkel (a reference worth citing) in their work investigating the corresponding states of attractive particles. Note that the L-L binodal (plotted in Supplemental Figure 1) is known from this work to become metastable with respect to the S-L binodal (not shown) when the range of attraction is shorter than $0.14a$. This is very close to the range of attraction reported in this work for the critical Casimir interaction, which is also longer range than other model attractive systems studied by Lu et al., Ramakrishnan et al., and others who have used the depletion interaction to induce attraction.

Thank you for this suggestion; we have added a remark about the mapping onto the adhesive hard sphere system, together with the reference by Noro and Frenkel. We added in the SI: “**This allows us in particular to compare our state points with those of the adhesive hard sphere system, which should map onto our short-range attractive system following the Noro-Frenkel correspondence [3]**”

We also added a remark concerning the solid-liquid binodal: “**We also note that the indicated fluid-fluid binodal in Figure S1 is known to become metastable with respect to the solid-liquid binodal (not shown) when the range of attraction is shorter than $0.14a$. This is close to the investigated attraction range of the critical Casimir interaction studied; however, we do not observe any sign of solid-liquid transition in the range of attractions we investigated.**”

The gel boundary is interesting to note as well. Varga and Swan report that the gel boundary for sticky spheres is sensitive to hydrodynamic interactions. At a volume fraction 0.12, the expected gel boundary from their work with far-field hydrodynamics is close to the experimental boundary reported by Campbell et al. (see Varga and Swan, 2016 for a discussion). In the absence of hydrodynamic interactions, as would be the case for the methods used here, the boundary is shifted to a higher attractive strength. I cannot understand this discrepancy.

Thank you for pointing us to this reference, which we have cited it in the revised manuscript. In their paper Varga and Swan, *Soft Matter* 12, 7670, 2016, the authors find this difference for colloidal systems with short-range attraction and long-range repulsion. Our repulsive and attractive length scales, in contrast, are similar, both of the order of 20nm, and have the same

distance dependence to first approximation (screened exponential). We indeed observe very similar structures in experiments and simulations (compare Figs. 1c (exp.) and 1f or SI Fig. S3c/d (sim.)), and in particular do not observe in our experiments any anisotropic cluster growth or alignment as predicted for the hydrodynamic simulations by Varga and Swan.

Along similar lines, how, then, does the critical transition of the lutidine mixture affect hydrodynamic interactions in the suspension? Are the near-field and / or far-field hydrodynamics altered? (I would assume so, given that the viscosity changes in the vicinity of the mixture phase separation. How does this affect the formation and breakage times of the colloidal "bonds"?)

We do not think that the critical Casimir interaction employed here adds anything unusual. Concerning the viscosity, we are still sufficiently far away from the critical point of the solvent mixture (in the current manuscript $\Delta T=0.5\text{K} \dots 1\text{K}$), where the “critical” contribution to the viscosity is unimportant, and the viscosity is dominated by the mean-field value. This is shown by the viscosity as a function of reduced temperature in Fig. A2 below, where we have highlighted the temperature range of our experiments in yellow. In this graph, taken from Stuij *et al.* *Soft Matter* **13**, 5233 (2017), we combine experimental and theoretical values, which show very good agreement with each other. In the highlighted range, the viscosity still behaves fine; even at 0.5K, there is no significant increase in viscosity. In fact, at these temperatures, the solvent correlation length is still $<20\text{nm}$, much smaller than the particle size, thus there is a large separation of scales between the fluctuation and colloidal size, meaning that the colloidal and solvent fluctuation dynamics decouple. This is in fact how we have designed the system initially. We thus can treat the colloidal dynamics as occurring in the presence of effective interactions, much more than much of the published work on colloidal depletion interaction, where this separation of scales is often not guaranteed.

Figure A2 Viscosity as a function of reduced temperature. Experimental measurements (red dots) are combined with theoretical estimates (green and blue lines). Data from Stuij *et al.*, *Soft Matter* (2017). The temperature range of the gelation experiments, $\Delta T=0.5 - 1\text{K}$ is highlighted in yellow (dashed box); no significant deviation from the normal temperature dependence is observed.

The critical value of the coordination number is reported as $z_c = 5.5$ for the experiments. It doesn't look like this is the same value in the simulations, but a value is not given. Instead, z_c for the simulations appears to be much lower. Why?

The final (steady-state) value of the average coordination number z in experiment and simulations is very similar, between 6.5 and 7.5. This also holds for all the other volume fractions that we have included in the revised version. The difference in the value of z at gelation (z_c) depends on the volume fraction as shown by Fig. A1a above (new Fig. S3a in the SI). Despite this system-specific difference, all curves collapse when scaling by z_c , as shown in Fig. A1b above (new Fig. S3b in the SI). Also, the scaling is robust (inset), supporting our main claim and showing its generality. We note that differences between experiments and simulations at the same volume fraction can arise due to the unavoidable presence of gravity (becoming significant for larger structures despite density matching), requiring larger z in experiment to be (marginally) stable at gelation. This difference vanishes in the later stages of gelation, when the gel structure has become more rigid. A more precise comparison of z at gelation would thus require experiments in microgravity.

The results don't really create a framework that unify structural arrest and yielding. This is highlighted late in the conclusions without any sort of empirical support by the current work. This would seem to imply, based on the assumptions of the model presented here, that yielding consists of some sort of surface erosion of the cluster, but perhaps I am misunderstanding their idea.

This seems indeed a misunderstanding, and we apologize for the lack of clarity in the original manuscript: the yielding does not refer to individual clusters, but to the yielding of a dense, bulk material. The idea is that when a solid (i.e. a macroscopic arrested material, e.g. a glass) is subjected to shear, flow starts at small clusters (nonaffine clusters, “shear transformation zones”) that grow until the largest cluster percolates and carries the flow, see Fig. A3, where we show only the most liquid-like particles in the otherwise dense, arrested glass. We indeed find in this work that the growth of nonaffine clusters exhibits a power law upon approaching the yield point, with exponent consistent with 3D percolation theory. This yielding process is the mirror image of what we find in the present gelation case, where, in an initial *fluid* phase, a *rigid* phase emerges, again starting from small local clusters that grow until the largest cluster percolates and the system acquires rigidity. Thus, while in yielding, percolation of a fluid happens inside a solid, in gelation, percolation of the solid happens inside the fluid phase; yet the percolation process seems to be similar. Thus, in a broad picture of flow-arrest transitions, the underlying nonequilibrium transitions appear to occur similarly, both based on a nonequilibrium percolation process, and this is what we wanted to highlight in a qualitative picture. We have better explained this analogy in the revised version, and hope the referee agrees with it. We rephrased: “**The yielding of amorphous solids has likewise been identified as nonequilibrium percolation transition [41,42]. Because this yielding process, which fluidizes an initially solid material can be regarded as a process opposite to gelation, which solidifies an initially fluid-like sample, it appears that the observed onset of rigidity from a fluid state, and the onset of flow from a solid state are two almost mirror-image manifestations of the same nonequilibrium continuous phenomenon.**”

If the referee, however, still finds this connection confusing, we'll be happy to remove it, but we find it an appealing qualitative picture that places our observations in a broader context.

Minor question: How can the entirety of the growing (largest) cluster be completely observed within the imaging volume?

Microscopy always observes a subsystem of an otherwise much larger (near-macroscopic) sample. In the experiment, we scan different areas of the sample to make sure that the observed field of view provides a representation of the bulk, but the field of view then is always limited to a subsystem, like in all other microscopy work. We have made this point clear in the revised version by modifying the sentence to “For each attraction, we follow the particle-scale aggregation process in a $108 \mu\text{m}$ by $108 \mu\text{m}$ by $40 \mu\text{m}$ volume using confocal microscopy.”

[Redacted]

Figure A3. Microscopic yielding of glasses: percolation of fluid-like clusters. The reconstructions highlight the highly nonaffine clusters in the otherwise dense amorphous structure (not shown) in a sheared colloidal glass. The nonaffine clusters grow with applied strain (a, 2.1%, b, 4.9%, and c, 10.1% strain) until the largest non-affine cluster percolates the field of view. Taken from our work Ghosh *et al.*, *Phys. Rev. Lett.* **118**, 148001 (2017).

REVIEWERS' COMMENTS:

Reviewer #1 (Remarks to the Author):

I am satisfied with the responses to my questions. The rewritten paper addresses all of my concerns and I am happy to recommend publication.

Reviewer #2 (Remarks to the Author):

The authors have sufficiently addressed my questions in their rebuttal and the revised manuscript. I have one further comment, and that is the distinction between the non-equilibrium and equilibrium percolation transition. There is a significant body of colloid research perhaps skirting this issue, such as that of Verduin and Dhont (H. Verduin, J. K. G. Dhont, *J Colloid Interface Sci.* 172, 425–437, 1995) and other studies in the rheology literature (Zukoski, Russel, etc.) It would be particularly useful if the authors explicitly addressed the differences between the current model and the assumptions relied on in the prior (and long-standing) literature.

Reviewer #2 (Remarks to the Author):

The authors have sufficiently addressed my questions in their rebuttal and the revised manuscript. I have one further comment, and that is the distinction between the non-equilibrium and equilibrium percolation transition. There is a significant body of colloid research perhaps skirting this issue, such as that of Verduin and Dhont (H. Verduin, J. K. G. Dhont, J Colloid Interface Sci. 172, 425–437, 1995) and other studies in the rheology literature (Zukoski, Russel, etc.) It would be particularly useful if the authors explicitly addressed the differences between the current model and the assumptions relied on in the prior (and long-standing) literature.

We thank the referee for their positive evaluation of our revised paper, and additional comment. Concerning the distinction between equilibrium and non-equilibrium transition:

We believe that (almost) all of the previous theoretical work addresses equilibrium percolation. The paper by Dhont is based on the well-known Chiew-Glandt theory of percolation in equilibrium adhesive hard sphere fluids, to which also the later papers by Norman Wagner (cited in the manuscript) refer. This theory uses static correlation functions of equilibrium fluids to predict the equilibrium percolation threshold as a function of volume fraction for different attraction. These theories do not predict critical exponents so they cannot be used to interpret our experimental and simulation data. Experimentally, some of the data indicated in the phase diagram (e.g. Fig. 14 of the Dhont paper), is dynamic data referring to dynamic arrest, but this data is very close to the phase separation line, looking rather related to the widely discussed dynamically arrested spinodal decomposition.

The paper by Russel is more focused on rheology, using static correlation functions to try to rationalize the scaling of the modulus $G' \sim (\phi - \phi_c)^s$, but to our knowledge there is no theory, to date, which can predict the scaling exponent of the modulus. Actually, the authors argue that the measured exponent $s \sim 3.0$ is not compatible with equilibrium percolation theory. Also here, the description of percolation is based on equilibrium static correlation functions, so once again everything is time-independent and equilibrium-like, and there is nothing about the cluster size distributions.

The work of Zukoski is also focused on the shear modulus of a colloidal gel, with theoretical work of Schweizer based on idealized MCT. This is closer to the nonequilibrium scenario, however the transition is seen as a sort of MCT transition of dense clusters, which is far away from our case of colloidal gelation at lower packing fraction, and also far from a second-order phase transition scenario that we see.

Thus, the early theory (late 80's and 90's) builds on the equilibrium Chiew-Glandt theory of percolation, to which most of the early experimental work refers, while the somewhat later theory efforts (2000's) refer to MCT, with experimental data based on rheology and/or light scattering being explained in terms of equilibrium percolation, dynamic arrest in terms of MCT, or arrested spinodal decomposition.

In our work, we take a different route: we show that percolation happens in a nonequilibrium process (as proven by the breaking of detailed balance), and we relate the directly observed diverging cluster sizes and critical exponents to a kinetic (i.e. time-dependent) cluster growth model, solved for the case of broken detailed balance. This establishes (for the first time to our

knowledge) a genuinely nonequilibrium percolation process with new critical exponents, both in experiments and modelling.

We have included the references suggested by the referee, and amended the corresponding sentence in the revised manuscript to:

“Yet, while equilibrium percolation transitions have been discussed for fluid-fluid and fluid-solid transitions [Chiew1983, Marr1993], used to interpret experimental data [Dhont95, Russel93, Eberle11], and in theoretical models [Broderix], their validity for systems out of equilibrium remains unclear”